# Validation of an Activity Type Recognition Model Classifying Daily Physical Behavior in Older Adults: The HAR70+ Model

**DOI:** 10.3390/s23052368

**Published:** 2023-02-21

**Authors:** Astrid Ustad, Aleksej Logacjov, Stine Øverengen Trollebø, Pernille Thingstad, Beatrix Vereijken, Kerstin Bach, Nina Skjæret Maroni

**Affiliations:** 1Department of Neuromedicine and Movement Science, Faculty of Medicine and Health Sciences, Norwegian University of Science and Technology, 7034 Trondheim, Norway; 2Department of Computer Science, Faculty of Information Technology and Electrical Engineering, Norwegian University of Science and Technology, 7034 Trondheim, Norway; 3Health and Care Services, The Municipality of Trondheim, 7004 Trondheim, Norway

**Keywords:** older adults, physical activity, daily physical behavior, accelerometer, walking aids, free-living, human activity recognition, machine learning

## Abstract

Activity monitoring combined with machine learning (ML) methods can contribute to detailed knowledge about daily physical behavior in older adults. The current study (1) evaluated the performance of an existing activity type recognition ML model (HARTH), based on data from healthy young adults, for classifying daily physical behavior in fit-to-frail older adults, (2) compared the performance with a ML model (HAR70+) that included training data from older adults, and (3) evaluated the ML models on older adults with and without walking aids. Eighteen older adults aged 70–95 years who ranged widely in physical function, including usage of walking aids, were equipped with a chest-mounted camera and two accelerometers during a semi-structured free-living protocol. Labeled accelerometer data from video analysis was used as ground truth for the classification of walking, standing, sitting, and lying identified by the ML models. Overall accuracy was high for both the HARTH model (91%) and the HAR70+ model (94%). The performance was lower for those using walking aids in both models, however, the overall accuracy improved from 87% to 93% in the HAR70+ model. The validated HAR70+ model contributes to more accurate classification of daily physical behavior in older adults that is essential for future research.

## 1. Introduction

Older adults differ widely in physical function, health, and level of physical activity [1]. Although advanced age is generally associated with a gradual decline in muscle strength, aerobic capacity, balance, and flexibility [2,3,4,5], the onset of decline varies among older adults and is conditioned by several individual factors [5,6], most notably level of physical activity, which is well established as an important determinant for maintaining function and independence in older age [1,7]. To be able to investigate the role of physical activity in relation to various health outcomes, there is a need for objective, valid tools to measure physical behavior in daily life. Activity monitoring using wearable sensors such as accelerometers is becoming the preferred method and is increasingly used in population studies to complement or replace self-reported measures of physical activity [8,9,10]. The most common approach for analyzing data derived from accelerometers has been to process the raw data into so-called activity counts, which classify activity based on predefined intensity-specific cut points [11,12]. However, classifying physical behavior according to commonly used cut points has been shown to underestimate physical activity in older adults [13]. Older age is related to lower cardiorespiratory fitness and higher metabolic cost of daily living activities [14,15]. Thus, everyday life activities, which are generally classified as low-intensity physical activity, can for many older adults represent physical behavior essential to maintaining physical function and independence [16]. Walking is the most popular physical activity type in adults [17] and is involved in nearly all everyday life activities for most individuals. Therefore, an alternative approach to cut points that may be more relevant in older adults is to classify types of daily physical behavior, including postures and activities related to mobility, such as standing and walking [18]. Furthermore, increasing age is related to lower gait speed [19,20], and walking events have been shown to be underestimated at slow gait speed by commercially available activity monitors that classify activity types [21].

Monitoring of physical activity combined with machine learning (ML) methods can contribute to a more accurate and detailed classification of daily physical behavior in older adults (e.g., [22,23,24,25]). However, existing ML models are typically developed and validated on training datasets collected from young, healthy adults [26], and it is unknown to what extent such models accurately classify daily physical behavior in older adults. Only a few studies have evaluated ML models for classification of daily physical behavior in older adults [23,27,28,29,30,31,32,33,34], and none of these evaluated the performance of ML models for older adults using walking aids. Older adults that use walking aids, such as a walker, represent a considerable sub-population of persons over 70 years of age [35]. Representing the population of interest is a foundational principle in validation studies and consequently, training datasets of older adults should include the variation over the entire span from fit to frail [18]. Furthermore, most validation datasets have been collected in controlled laboratory settings or with highly structured protocols [21,28,29,30,31,34,36]. Two previous studies that evaluated the accuracy of laboratory-based ML models classifying activity from free-living settings in older adults showed that the transferability was low and hence, that free-living datasets are necessary to develop accurate models [32,36]. Data collection in free-living settings is more challenging than in laboratory settings, which might explain why few validation studies are conducted in real-life conditions with older adults. Additionally, only three of the earlier studies used video recordings as ground truth in free-living settings [23,28,33], which is regarded as the best reference measure in validation studies [18]. 

A recently published study for classification of daily physical behavior from a dual accelerometer setup showed promising results, with an overall accuracy of 96% in detecting six activity types (sitting, lying, standing, walking, running, and cycling) in free-living data from healthy adults with an activity type recognition model [22]. We wanted to investigate how generalizable this model is for a sample aged above 70 years, and thereby evaluate the need for age-specific models. Hence, the specific aims of the current study were to: (1) evaluate the performance of an existing activity type recognition ML model for the classification of daily physical behavior in a sample of fit-to-frail older adults, (2) compare the performance with an updated ML model that includes training data from the sample dataset of older adults collected in this study, and (3) evaluate the performance of the ML models for those using walking aids compared to those not using walking aids.

## 2. Materials and Methods

### 2.1. Participants

To be included, participants had to be aged 70 years or older and be able to walk independently with or without walking aids. Eighteen older adults aged between 70 and 95 years were included in the data collection. Characteristics of the participants are presented in Table 1. Four participants used a walker during all walking activities and one participant used walking sticks when walking outdoors. Participants were recruited within a municipality in Norway, in collaboration with physiotherapists and through acquaintances of one of the researchers and other participants. All participants provided informed, written consent. The data collection took place from April until June 2021. The study was approved by the Norwegian Centre for Research Data and conducted in accordance with the Declaration of Helsinki.

### 2.2. Procedure

The semi-structured validation protocol took place in and nearby the participants’ own homes. They were equipped with two accelerometers and a chest-mounted camera and performed several repetitions of the daily living activities walking, standing, sitting, and lying. Age (years), sex, and self-reported height and weight were registered. The Human Activity Recognition 70+ (HAR70+) dataset from this study is publicly available for further development and to enable objective comparisons between models in future research (https://github.com/ntnu-ai-lab/harth-ml-experiments/tree/v1.2 (accessed on 19 February 2023)).

#### 2.2.1. Activity Monitors

Participants were fitted with two Axivity AX3 accelerometers (Axivity, Newcastle, UK) positioned on the lower back (centrally at the third lumbar vertebrae) and the right thigh (front midline approximately 10 cm above the upper border of the patella) directly to the skin using Opsite Flexifix transparent film (Smith & Nephew, Watford, UK). The accelerometers’ placement is illustrated in Figure 1. The AX3 is a lightweight (11 g, 23 × 32.5 × 7.6 mm) triaxial accelerometer that records movement data at high frequency for up to 14 days at 100 Hz. The accelerometers were oriented with the USB port (positive *x*-axis) pointing downwards and the fabric print visible (positive *z*-axis towards the skin). The AX3 GUI software (version 1.0.0.43) was used to configure the accelerometers to record with a sampling frequency of 50 Hz and range of ±8 g. The AX3 and its software are open source and, hence, allow for data processing of raw accelerometer data.

#### 2.2.2. Video Recordings

A GoPro Hero 8 camera (San Mateo, CA, USA) was attached to the chest of the participants with a chest harness (GoPro Chest Mount Harness). The camera was pointing downwards filming only the abdomen and lower limbs of the participants. Video recordings allow for identification of activities based on lower body movements and the orientation of the body relative to the surroundings. Video files were recorded with a resolution of 1920 × 1080 pixels at 60 frames per second (fps) and were stored in MP4 format.

#### 2.2.3. Validation Protocol

After being fitted with the accelerometers and camera, the participants were instructed to perform three repetitions of heel drops, meaning that they raised up on their toes and dropped the heels on the ground, to synchronize the two accelerometers and the video recording. For the participants that were not able to perform this procedure, the researcher clapped the accelerometers together three times in front of the camera before attaching them to the participant’s back and thigh. In this way, reference points that were easy to identify for synchronization afterwards were obtained. Then, a semi-structured free-living protocol for classification of physical behavior was carried out. The protocol included the activity types walking, standing, sitting, and lying. One researcher ensured that the protocol included a minimum of 5 min of standing, sitting, and lying, and a minimum of 15 min of walking. All activity types were distributed over several bouts. The first part of the validation protocol took place inside the home of the participants (approximately 25 min), and the second part took place outside (approximately 15 min). The researcher focused on keeping the behaviors as natural as possible to avoid a “laboratory feeling” and encouraged the participants to change between activity types using simple instructions. For example, they could be instructed to go to the kitchen and fill a glass of water. Their natural surroundings were used to obtain variation that was representative for each participant. For instance, the participants were first instructed to sit down where they usually sit during a normal day. If this was in a chair, they could be instructed to sit down on their couch for the next sitting session. When walking outside, the researcher walked alongside the participant and gave instructions to create variation in the duration of walking bouts and encouraged them to vary walking speed to capture the variation existing within and across the subjects.

### 2.3. Data Processing

#### 2.3.1. Video Analysis

The video analysis was used as the ground truth for the validation of the activity types identified by the ML model. Predetermined definitions with exact descriptions of the onset and offset of activities were used for the video classification (Table 2). The labeled activities were walking, stair walking (up/down), shuffling, standing, sitting, and lying. Walking was defined as locomotion towards a destination with one stride (i.e., two steps) or more. The shuffling category represents periods with feet movements that were not classified as walking. The participants were not instructed to perform shuffling or stair walking; however, such movements are common activity types that occur during a free-living protocol and were thereby classified according to the definitions.

The video files were converted to 25 fps and a resolution of 640 × 360 pixels for the video analysis. Labeling was done frame-by-frame according to the predetermined activity definitions using the Anvil video annotation tool (version 6.0) [37]. The synchronization periods in the beginning and end were identified, and video recordings were labeled in full length between these two periods and independently from the accelerometer data. The video recordings were labeled by one of two raters following the coding described in detail here: https://github.com/ntnu-ai-lab/harth-ml-experiments (accessed on 19 December 2022). In addition, individually labeled video recordings from four participants were randomly chosen to test for inter-rater reliability of the video analysis.

#### 2.3.2. Accelerometer Data Pre-Processing

The same data pre-processing steps as described in Bach et al. 2021 [22] were applied in this work. First, the raw data were downloaded from the internal memory of the accelerometer as a binary file (Continuous Wave Accelerometer format) using the AX3 GUI software. Second, a fourth-order Butterworth band-pass filter was applied to all six signals (two sensors, three axes). Third, the back and thigh sensors, as well as the labeled data, were synchronized and stored as a CSV file. Afterwards, the signals were segmented into non-overlapping, five-second time windows (250 samples at 50 Hz). This allowed the extraction of features for each segment, which can be used to train the ML classifier. The same 161 features as in the work of Logacjov et al. 2021 [38] were computed. The 10 features that provide the most information for activity type prediction in the two models are shown in Figure 2. For each five-second segment, the dominant activity type labeled within that segment was used as ground truth for ML.

#### 2.3.3. Machine Learning

Bach et al. 2021 [22] trained an extreme gradient boost (XGB) model [39] to perform activity type recognition on the Human Activity Recognition Trondheim (HARTH) dataset [38], which contains labeled accelerometer data of healthy young adults. XGB is an ensemble learning technique that consists of multiple, sequentially aligned weak classifiers, namely decision trees. Each decision tree is trained to solve the previous decision trees’ mistakes by minimizing a loss function. The weighted sum of each weak classifier in the XGB model constitutes the final activity type prediction [39].

Two ML models were evaluated in the current work. First, the experiments of Bach et al. [22] were reproduced by training the XGB approach on the newest version of the HARTH dataset. As in the original work, a grid search with cross-validation was performed to find optimal model hyperparameter assignments. The F1 score was used to evaluate the performances. The best model configuration had a learning rate of 0.5, a maximum decision tree depth of 3 for 1024 decision trees in total, as well as regularization parameters λ = 1 and α = 0. The multi-class classification error rate was used as the loss function. The best model configuration was then trained on the entire HARTH dataset. The resulting model is referred to as the HARTH model in this work as it is a reproduced version of an existing XGB. Subsequently, the HAR70+ dataset was used to evaluate how the HARTH model, trained on young adults only, performs in older adults.

Second, the HARTH and the HAR70+ datasets were combined to train a second XGB. Again, a grid search with cross-validation was executed for hyperparameter optimalization. The best model configuration had a learning rate of 0.3, a maximum tree depth of 5, a decision tree number of 1024, and regularization parameters λ = 1 and α = 0. Afterwards, this configuration was trained in a leave-one-subject-out cross-validation (LOSO). In a LOSO, the ML model is trained on N-1 subjects and tested on the remaining one. N is the total number of subjects in the dataset; hence, N = 40 for HARTH and HAR70+ combined. The training was repeated N times such that each subject was used for testing once. A LOSO allows the investigation of the model’s performance on older adults’ data it was not trained on, making it comparable to the HARTH model’s results. The resulting model is referred to as the HAR70+ model.

The target activity types in this study were walking, standing, sitting, and lying. We combined classes for the analyses so that stair walking categories were integrated with the walking category and shuffling was integrated with standing. Although the data collected in this study did not include the activity types running and cycling, the ML models were additionally trained on these categories, and hence these activity types are also classified [22].

### 2.4. Statistical Analysis

To assess the inter-rater reliability of the video analysis, Cohen´s kappa statistic was used [40]. Furthermore, agreement between the labeled accelerometer data and the predicted ML data was assessed by comparing each sample and generating confusion matrixes. As an estimate of the overall performance, overall accuracy was calculated for each model by dividing the proportion of correctly classified samples by the total number of samples. Because of the imbalance in class distribution (Figure 3), Kappa statistics were calculated as an additional estimate of the overall performance for each model to measure the level of agreement beyond that expected by chance [40]. Sensitivity (proportion of true positives correctly identified), specificity (proportion of true negatives correctly identified), and precision (proportion of true positives out of true and false positives) were calculated as means with a 95% confidence interval across participants for each of the activity types walking, standing, sitting, and lying. The F1 score was calculated as the harmonic mean between sensitivity and precision with the formula:(1)$$F1\;\mathrm{score}=2\times \frac{\mathrm{Sensitivity}\;\times \;\mathrm{Precision}}{\mathrm{Sensitivity}\;+\;\mathrm{Precision}}$$

The F1 score, sensitivity, specificity, and precision all range between 0 and 1, and the closer to 1, the more precise the model. All statistical analyses were performed using STATA Statistical Software version 17 and Microsoft Excel.

## 3. Results

### 3.1. The HAR70+ Dataset

The total duration of labeled video data was 12.6 h (mean 41.8 ± 5.8 min). The inter-rater agreement between the two independent raters was high, achieving a Kappa value of 0.95. All participants completed the full protocol, except one participant that did not perform lying due to fear of causing dizziness. The time distribution of the seven labeled subcategories and the merging into four activity types are presented in Figure 3. The mean time per subject for the activity types was 20.2 ± 3.9 min for walking, 9.0 ± 3.0 min for standing, 8.9 ± 3.3 min for sitting, and 3.8 ± 1.6 min for lying. The participants’ number of bouts for each activity type was on average 46.2 ± 17.2 for walking, 84.8 ± 28.9 for standing, 51.1 ± 19.5 for shuffling, 4.2 ± 1.6 for sitting, and 1.2 ± 0.5 for lying. Thirteen participants had at least one bout of stair walking with a mean duration of 4.1 ± 3.0 s.

### 3.2. Performance of the ML Models

The overall accuracy classifying the activity types walking, standing, sitting, and lying was 91% (Kappa 0.87) for the HARTH model in fit-to-frail older adults. For those using walking aids, overall accuracy was 87% (Kappa 0.82) while for those not using walking aids the overall accuracy was 92% (Kappa 0.89). The HAR70+ model which included training data from older adults had a slightly higher overall accuracy of 94% (Kappa 0.92). Furthermore, for the HAR70+ model, the overall accuracy was 93% (Kappa 0.90) and 95% (Kappa 0.93) for those walking with and without walking aids, respectively.

The performance of the ML models in classifying daily physical behavior is presented in Table 3, and Figure 4 shows the corresponding confusion matrixes. For both models, misclassifications mainly occurred between the activity types walking and standing, and between sitting and lying (Figure 4). Walking and standing were classified with high precision in both ML models, with the HAR70+ model showing a slightly higher F1 score for walking (0.95 versus 0.93) and standing (0.89 versus 0.86) compared to the HARTH model. The HARTH model had a sensitivity of 0.93 and precision of 0.79 for standing, compared to 0.87 and 0.90, respectively, for the HAR70+ model. For walking, the sensitivity and precision for the HARTH model were 0.88 and 0.97, respectively, compared to 0.95 and 0.94 for the HAR70+ model, indicating that the HARTH model more often misclassified walking as standing and that the HAR70+ model more often misclassified standing as walking, when compared to each other. Furthermore, the HAR70+ model predicted sitting and lying with considerably higher precision compared to the HARTH model (Figure 4). The HARTH model misclassified more than 20% of lying as sitting, while for the HAR70+ model, sitting and lying were almost perfectly predicted with F1 scores of 0.98 for sitting and 0.96 for lying (Figure 4).

For both models, the performance of classifying daily physical behavior was lower for those using walking aids compared to those who did not. The higher performance for the HAR70+ model in the group of older adults using walking aids explained much of the higher overall accuracy seen in total for the HAR70+ model compared to the HARTH model (Table 3).

Even though the HAR70+ training dataset did not include the activity types running and cycling, the ML models were trained on these activity types and were classifying them. Correctly, running was not classified at all in this sample dataset. However, 23 five-second windows of cycling that were actually walking were misclassified by the HARTH model. Three of these windows appeared as single cases while a long bout of 20 consecutive five-second windows occurred in a participant walking with walking aids. For the HAR70+ model, only two single cases of five-second windows of walking were misclassified as cycling.

## 4. Discussion

This study evaluated the performance of a new activity type recognition ML model (HAR70+) that included training data from older adults for classification of daily physical behavior in a sample of fit-to-frail older adults and compared the performance with an earlier ML model (HARTH). The results showed that daily physical behavior can be classified with high accuracy in older adults using a dual accelerometer setup and machine learning. Although the HARTH model was trained on data from young healthy adults only, it classified the four activity types evaluated in this study with high accuracy. However, the overall accuracy improved from 91% to 94% for the HAR70+ model after including data from older adults in the training dataset.

An overall purpose of activity monitoring is to be able to classify physical behavior in older adults in their daily environment with high precision. However, the accuracy of ML models for classification of free-living physical behavior can be compromised if the characteristics of the population of interest are not adequately represented in training datasets [18]. Therefore, we made a specific effort to cover the variation in the entire range from fit to frail in the sample dataset in the current study. When it comes to characteristics of physical behavior, frail older adults using walking aids are likely to differ the most from younger adults, due to the leaning-forward posture while using a walker, decreased swing and stance times during walking that often results in slower walking speed, and an overall different gait pattern [41]. The importance of including training data from individuals using walking aids was reflected in the results which showed that the existing HARTH model classified physical behavior with lower accuracy in those using walking aids compared to those who did not (overall accuracy: 87% versus 92%), and that the difference was reduced in the new HAR70+ model (overall accuracy: 93% versus 95%). To the best of our knowledge, only three previous studies have included older adults using walking aids in data collection for activity type recognition [27,28,33], but none of these separately reported the performance of the ML models for the participants using walking aids. In the current study, we evaluated the performance of both ML models for the frailest participants using walking aids compared to the participants with better physical function not depending on a walking aid during walking. Our results indicate that the new HAR70+ model, developed and evaluated in the current study, should be used to accurately describe daily life physical behavior for older adults that may use walking aids.

The HAR70+ model was considerably more precise in classifying sedentary behavior than the HARTH model. The HARTH model misclassified more than 20% of lying as sitting, while these activity types were almost perfectly classified in the HAR70+ model. The better prediction of sitting and lying in the HAR70+ model compared to the HARTH model explains much of the higher overall accuracy in the model trained on data from older adults. In the current dataset we found that misclassification of lying appeared when an older person had a small angle in their upper body, typically when resting their head on the sofa arm. This demonstrates the importance of obtaining data on physical behavior in a realistic environment, such as the participants’ own homes, as we did in the current study. Additional training data from older adults also captured more of the variation between different people, which also may have contributed to a more robust classification model.

The performance of classification of walking and standing was high in both ML models, but the models differed in which activity type they misclassified the most. The HARTH model mostly misclassified walking as standing when tested on the sample dataset of older adults, while it was the other way around for the HAR70+ model. The overall performance for walking and standing (F1 score) improved marginally in the HAR70+ model, indicating that this model performs better when classifying upright physical behavior in older adults. It is crucial to accurately classify upright physical behavior because walking and standing are involved in nearly all daily life activities for most individuals. The validated HAR70+ model presented here did not underestimate walking in older adults, which was reported in a previous study [21], and can therefore be applied to classify walking from activity monitoring in older adults with high accuracy. 

Even though the models predict running and cycling, the HAR70+ dataset did not include these activity types. Nevertheless, a few bouts of cycling were misclassified by the models. Although the impact on overall accuracy was small (<2 min for the HARTH model, and 10 s for the HAR70+ model out of 12.6 h labeled data), such misclassifications are important to detect because they can lead to significant misclassification in prolonged activity monitoring. For instance, for one participant walking with a walker, a 1 min and 40 s long bout was misclassified by the HARTH model as cycling. We presume that this was due to the accelerometer data having common features with cycling when the participant was leaning forward towards the walker combined with a low frequency of stepping. However, the same period was correctly classified as walking by the HAR70+ model, indicating that the latter model, which was trained on additional data from older adults, experiences less misclassification into cycling and therefore will be more robust also in older adults using walking aids.

Datasets collected in laboratory settings or with highly structured protocols often lack the sporadic transitions between activities that are characteristic of free-living behavior, which might explain why previous studies have shown low transferability between laboratory and free-living settings [32,36]. We made a conscious effort to collect the data for the HAR70+ dataset in a natural environment so that the actual settings of the physical behavior would be represented in the validation dataset [18]. The data collection in this study took place in and nearby the participants´ own homes where they performed everyday activities, with numerous repetitions of walking, standing, sitting, and lying. Typical bout length of walking inside a home environment is expected to be shorter than typical bout length outside [42], and it was therefore important to include both settings to represent the variation seen in free-living behavior [43]. The HAR70+ dataset had a high number of walking and standing bouts with short durations, which is typical in daily life [43]. The HAR70+ model showed high performance in discriminating between walking, standing, and sedentary behaviors despite frequent transitions between the activities. Therefore, the validated HAR70+ model is recommended when classifying daily physical behavior in older adults from free-living datasets.

The ML models evaluated in the current work extracted 161 features for each five-second segment of the dataset. The 10 most important features for activity type prediction (presented in Figure 2) showed that the two models differed in which features were the most critical. The difference in feature importance between the HARTH and HAR70+ models emphasizes that age and function impact the feature selection and thus, that there is a need for age- and population-specific models.

A few earlier studies have evaluated ML models for classification of daily physical behavior in older adults as well [23,27,28,29,30,31,32,33,34], but comparing results across the studies is challenging because of the large variety in methodology, such as the study population, validation protocol, reference measures, data analysis, and outcome variables, as well as type, number, and placement of activity monitors [26]. The study conducted by Awais et al. (2019) had many similarities to the current study, where they classified walking, standing, sitting, and lying with an overall F1 score of 87% in older adults in free-living settings using two accelerometers [23]. Likewise, Taylor et al. (2014) found an overall accuracy of above 85% for walking, sitting, and lying, and 56% for standing using one accelerometer on the lower back [28]. Although a single accelerometer setup can be more convenient, we argue that a dual accelerometer setup is preferable due to its better capability to discriminate between sitting, lying, and standing positions [22,24]. A single accelerometer placed at the lower back does not accurately discriminate between sitting and standing because of the similar orientation of the accelerometer in these two activity types [22,24,28]. Moreover, standing is an activity type characterized by little movement and thereby minimal acceleration measured by an accelerometer. For that reason, the traditionally used cut points approaches might categorize standing as sedentary behavior. We claim that it is critical to be able to discriminate standing from sedentary behavior when monitoring activity because standing is fundamentally different from sitting and might represent an important activity in older adults’ daily life. Standing can have positive effects on functional abilities related to strength and balance, which in turn are related to maintaining independence [16]. Furthermore, being able to accurately discriminate between sedentary behavior and the upright activities standing and walking is crucial when evaluating interventions aiming to reduce sedentary time.

The major strength of the current study is the inclusion of fit-to-frail older adults, including participants using walking aids, in a free-living training dataset for classification of physical behavior using a dual accelerometer setup and machine learning. Moreover, the current study used a high-quality reference measure with video recordings as ground truth, which was labeled manually frame-by-frame using predefined activity definitions, showing high inter-rater reliability between two observers.

A potential study limitation may be the short recording length and the lack of a standardized data collection part conducted in controlled laboratory settings in older adults. In the current study, we focused on collecting data from older adults´ natural surroundings. Furthermore, classifying physical behavior into activity types does not account for the intensity of the performed activity. However, correctly classifying activity types is a necessary first step, and further development of ML models can involve classification of intensity, such as speed of walking activity.

## 5. Conclusions

This study presented the development and validation of a new activity type recognition ML model that can accurately classify daily physical behavior performed in free-living settings using a dual accelerometer setup in older adults. Although the previous HARTH model, based on data from young healthy adults only, performed well also for a sample of older adults, the accuracy improved with the new HAR70+ model, especially for the frailest participants using walking aids. The role of physical activity as an important determinant of healthy ageing is well established, but precise monitoring of physical behavior in daily life has remained challenging. The validated HAR70+ model represents an important step forward to accurately describe physical behavior in older adults, despite being a heterogeneous group with wide-ranging functional levels. Accurate description of daily life physical behavior is crucial in scenarios such as determining the level of physical activity, describing compliance with physical activity guidelines, evaluating interventions, or describing physical activity in relation to health or other factors. Thus, the new HAR70+ model for fit-to-frail older adults will contribute to more accurate and detailed knowledge of daily physical behavior that is essential for future research.

## Figures and Tables

**Figure 1 sensors-23-02368-f001:**
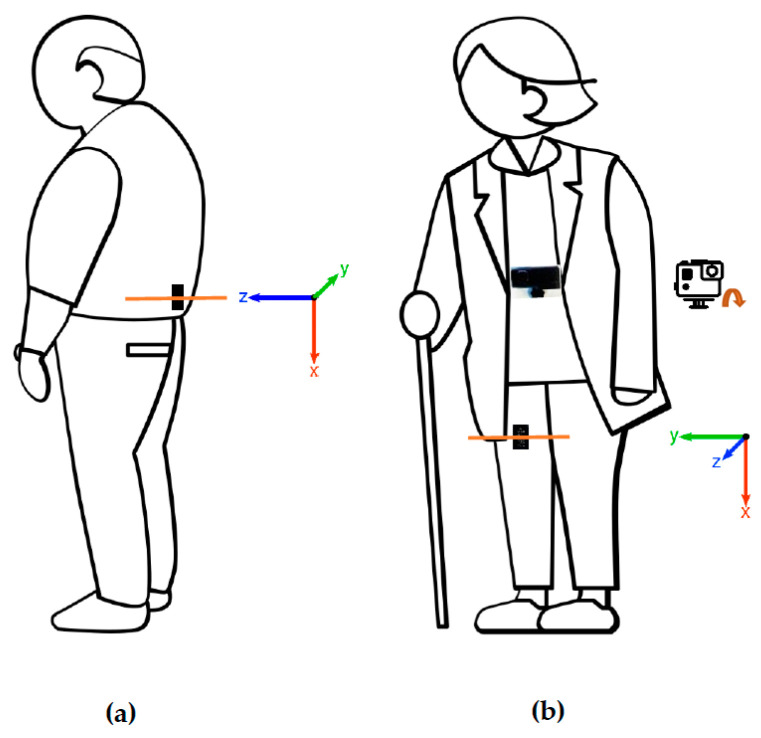
This figure illustrates the positions of the two accelerometers attached to the skin (highlighted with orange lines) and the chest-mounted camera used for the HAR70+ dataset. (**a**) The back accelerometer was positioned centrally at the third lumbar vertebrae. The *z*-axis of the coordinate system was pointing forward. (**b**) The thigh accelerometer was positioned approximately 10 cm above the upper border of the patella. The *z*-axis was pointing backward. The camera was attached to the chest on the outside of the clothing, pointing downwards.

**Figure 2 sensors-23-02368-f002:**
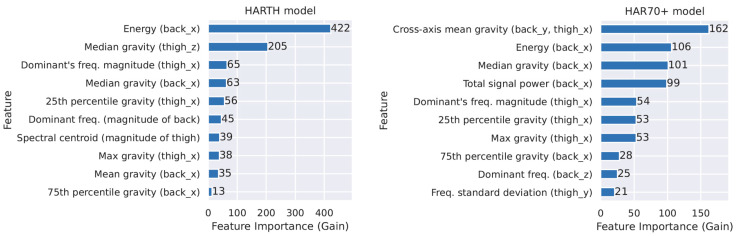
The 10 features that provide the most information for activity type prediction in the HARTH model (**left**) and the HAR70+ model (**right**).

**Figure 3 sensors-23-02368-f003:**
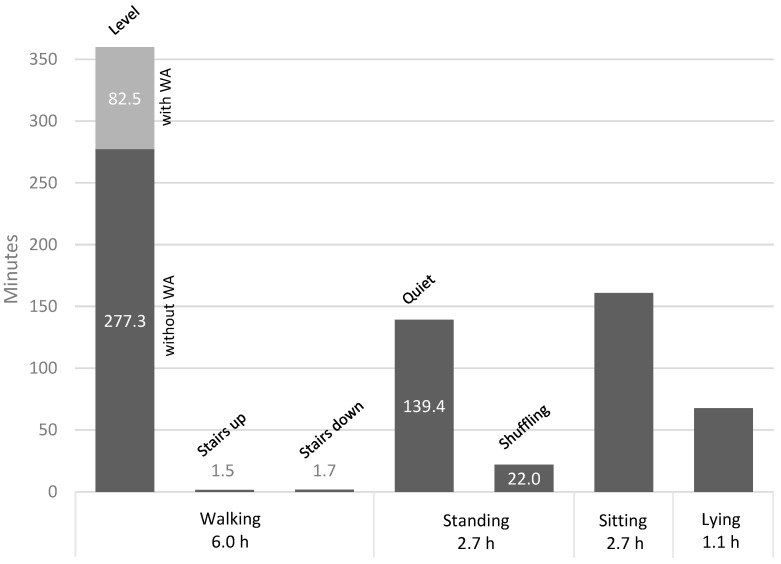
Time distribution of the labeled video data for the activity types during the semi-structured protocol. Light grey color represents the time of level walking with walking aids (WA).

**Figure 4 sensors-23-02368-f004:**
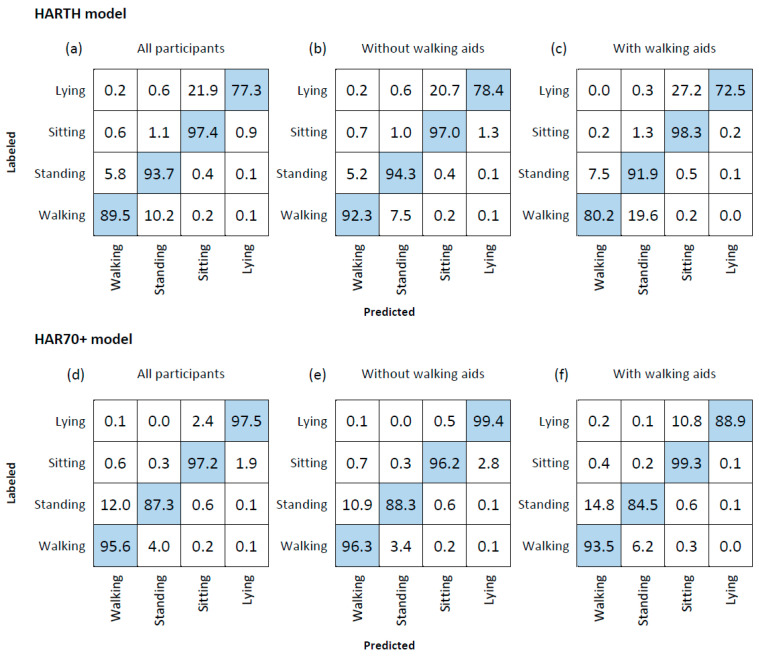
Confusion matrixes for the Human Activity Recognition Trondheim (HARTH) model (**a**–**c**) and the Human Activity Recognition 70+ (HAR70+) model (**d**–**f**) for all participants (n = 18) and separated for participants not using walking aids (n = 13) (**b**,**e**), and participants using walking aids (n = 5) (**c**,**f**). The rows represent the labeled activity types, while the columns represent predicted activity types. Values are shown in row percentages.

**Table 1 sensors-23-02368-t001:** General characteristics of the participants.

	Total Group	Without Walking Aids	With Walking Aids
n (%)	18 (100)	13 (72)	5 (28)
female (%)	9 (50)	6 (46)	3 (60)
age (years)	79.6 ± 7.6 [70–95]	77.2 ± 6.6 [70–95]	85.8 ± 7.0 [76–94]
weight (kg)	80.0 ± 9.3 [67–100]	79.8 ± 9.9 [67–100]	80.4 ± 8.8 [70–93]
height (cm)	173 ± 7.8 [161–186]	173 ± 8.0 [161–186]	171 ± 7.6 [164–180]
BMI (kg/m^2^)	26.8 ± 2.7 [22.6–31.2]	26.6 ± 2.8 [22.6–31.1]	27.6 ± 2.6 [25.1–31.2]

Note. Values are presented as mean ± SD and range. BMI: body mass index.

**Table 2 sensors-23-02368-t002:** The activity type definitions during video analysis.

Activity Type	Definition
Walking	Locomotion towards a destination with one stride or more (one step with each foot). Walking could occur in all directions. Walking along a curved line is allowed. Walking could occur using walking aids.
Stairs (up/down)	Start: Heel-off of the foot that will land on the first step of the stairs. End: When the heel-strike of the last foot is placed on flat ground. If both feet rests at the same step with no feet movement, standing should be labeled.
Shuffling	Stepping in place by non-cyclical and non-directional movement of the feet. Includes turning on the spot with feet movements that are not part of a walking bout. When not able to see the feet, and movement of the upper body and surroundings indicate non-directional feet movement, shuffling should be labeled.
Standing	Upright, feet supporting the person’s body weight, with no feet movement, otherwise, this should be labeled as shuffling/walking. Movement of upper body and arms is allowed. If feet position is equal before and after upper body movement, standing should be labeled. When not able to see the feet, and upper body and surroundings indicate no feet movement, standing should be labeled.
Sitting	When the person’s buttocks are on the seat of a chair, bed, or floor. Sitting can include some movement in the upper body and legs; this should not be labeled as a separate transition. Adjustment of sitting position is allowed.
Lying	The person lies either on the stomach, on the back, or on the right/left shoulder. Movement of arms, feet, and head is allowed.

**Table 3 sensors-23-02368-t003:** Sensitivity, specificity, and precision, calculated as mean [95% confidence interval] across participants, and F1 score, calculated as harmonic mean between sensitivity and precision, for classification of daily physical behavior for the Human Activity Recognition Trondheim (HARTH) model and the Human Activity Recognition 70+ (HAR70+) model.

	**HARTH Model**						
**Activity Type**	**Sensitivity**		**Specificity**		**Precision**		**F1 Score**
Total group							
Walking	0.88	[0.84, 0.92]	0.97	[0.97, 0.98]	0.97	[0.96, 0.97]	0.93
Standing	0.93	[0.92, 0.95]	0.93	[0.91, 0.95]	0.79	[0.76, 0.83]	0.86
Sitting	0.97	[0.96, 0.99]	0.97	[0.95, 1.00]	0.93	[0.87, 0.99]	0.94
Lying	0.79	[0.60, 0.99]	1.00	[0.99, 1.00]	0.86	[0.69, 1.00]	0.86
Without walking aids							
Walking	0.92	[0.91, 0.94]	0.97	[0.97, 0.98]	0.97	[0.97, 0.98]	0.95
Standing	0.94	[0.92, 0.95]	0.95	[0.94, 0.96]	0.83	[0.80, 0.86]	0.89
Sitting	0.97	[0.95, 1.00]	0.97	[0.94, 1.00]	0.93	[0.85, 1.00]	0.93
Lying	0.83	[0.61, 1.00]	1.00	[0.99, 1.00]	0.90	[0.73, 1.00]	0.86
With walking aids							
Walking	0.77	[0.66, 0.89]	0.97	[0.95, 0.99]	0.95	[0.93, 0.97]	0.86
Standing	0.92	[0.86, 0.98]	0.88	[0.81, 0.95]	0.70	[0.63, 0.78]	0.79
Sitting	0.98	[0.96, 1.00]	0.97	[0.92, 1.00]	0.93	[0.80, 1.00]	0.96
Lying	0.68	[0.00, 1.00]	1.00	[1.00, 1.00]	0.74	[0.00, 1.00]	0.84
	**HAR70+ Model**						
**Activity Type**	**Sensitivity**		**Specificity**		**Precision**		**F1 score**
Total group							
Walking	0.95	[0.94, 0.97]	0.95	[0.94, 0.96]	0.94	[0.93, 0.95]	0.95
Standing	0.87	[0.85, 0.89]	0.97	[0.97, 0.98]	0.90	[0.88, 0.92]	0.89
Sitting	0.97	[0.93, 1.00]	0.99	[0.99, 1.00]	0.98	[0.97, 0.99]	0.98
Lying	0.98	[0.94, 1.00]	0.99	[0.98, 1.00]	0.97	[0.91, 1.00]	0.96
Without walking aids							
Walking	0.96	[0.95, 0.97]	0.95	[0.94, 0.96]	0.95	[0.94, 0.96]	0.96
Standing	0.88	[0.86, 0.90]	0.98	[0.97, 0.98]	0.91	[0.89, 0.93]	0.90
Sitting	0.96	[0.91, 1.00]	1.00	[1.00, 1.00]	0.98	[0.98, 0.99]	0.97
Lying	0.99	[0.99, 1.00]	0.99	[0.98, 1.00]	0.96	[0.89, 1.00]	0.97
With walking aids							
Walking	0.93	[0.88, 0.97]	0.94	[0.91, 0.97]	0.92	[0.89, 0.95]	0.93
Standing	0.85	[0.78, 0.91]	0.96	[0.94, 0.98]	0.87	[0.81, 0.92]	0.86
Sitting	0.99	[0.98, 1.00]	0.98	[0.95, 1.00]	0.97	[0.91, 1.00]	0.98
Lying	0.91	[0.72, 1.00]	1.00	[1.00, 1.00]	0.99	[0.98, 1.00]	0.94

## Data Availability

The HARTH dataset presented in this article is available on https://github.com/ntnu-ai-lab/harth-ml-experiments/tree/v1.2 (accessed on 22 November 2022). The HAR70+ dataset will be available upon publication.

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
