# Peer review of "Validation of an Activity Type Recognition Model Classifying Daily Physical Behavior in Older Adults: The HAR70+ Model"

_sensors, 2023, doi:10.3390/s23052368_

Round 1

Reviewer 1 Report

Detailed studies on daily physical behavior in adults (elderly) are always interesting considering the continuous increase of this sector of the population and even a study of the impact before and after the pandemic would be interesting.

There is a significant number of articles on the subject that are not cited by the authors, I suggest improving the bibliographical research...

A table with comparative data from studies carried out by other authors would be useful, this includes methods, metrics, and tools used and how the proposed study stands out over these...

Inserting Table 1 as an appendix is not a good choice...

I suggest working a little more on formatting and describing the study... presentation of metrics (tables), patient classification criteria (tables)... presenting comparative data (existing in the literature)...

Bibliographic references need to be properly formatted and numbered according to the citation model used.

Reviewer 2 Report

Visualization of the results is clear and easy to interpret, text is well-written and easy to follow. Topic and content are important for scientific community, and new data sets are always of great value, especially bringing unique data, which is the case in this study - there were very few occasions to find mistakes in the text – my complements to the authors for professional editing, high language quality and clarity of writing

While I’m very positive about this study, I would like it to be more complete. It should be noted that this study is incremental approach to previous studies, the novelty lies mostly in the new data set, and comparison of new data set and older one using same ML model. Therefore, there are a couple of details that would be good to supplement to this work:

1.      Information about features that are used in the ML model to characterize data samples.

2.       Summary of features influence on the training results.

3.       Explain if anthropometric parameters (weight, height) and gender and age were used as features and if yes what are their influence on the classification results.

Below I summarized detailed comments for the text.

Introduction lines 36-37: “…most notably level of physical activity, which is well established as an important determinant of successful ageing” – I’m a bit surprised to see the phrase “successful ageing”, to me it sounds a bit strange in this context, I would use phrase like “longevity” or “healthy ageing”.

2.2 Procedure – “…self-reported height and weight were registered…” – while those parameters are not likely to influence interpretation of data and classification in a substantial way, I think that it is a reasonable effort to measure those quantities in lab setting, just to be sure that the registered values are measured in the same way (for example in underwear and not with full clothing that contributes up to 2kg of the measured body mass) and with known accuracy. Self-measured height by older individuals could be rather old data that is no longer up to date due to changes in posture. The range of error of self-made measurements is difficult to estimate, so, when possible, it should be avoided. This is just a remark for future studies, that while good amount of effort was spent to design good measurement protocol, simple measurements were left as subject input and not validated.

Section 2.2.1: while the description of accelerometer placement is quite sufficient, I think that adding a simple figure showing schematically sensors on the subject’s body and their orientation would be a good addition.

Section 2.3.2, lines 185-186: “The same 161 features as in the work of Logacjov et al. 2021 [31] were computed.” – This is one of the really interesting parts of the research, as feature selection has major impact on the model prediction performance. Nevertheless, I could not find any section in this paper that would describe which of those 161 features were the most important in terms of activity prediction. I understand that describing all those features in the current paper would be too much to ask for, but 10 that give the most information could be indicated. This would significantly strengthen the impact of this work.

Reference 1, line 472: line break should be inserted at the end of the first line of this reference as it looks awkward being justified with such large spaces between words, or the link should be split in another place so that the first portion of the link would still be in the first line of the reference. It is usually more readable if such long links are split on key/value of the query or slash being part of the path then in arbitrary position of the link. The actual link contains unnecessary session and sequence parameters, that bring nothing to the reference, except making it long, the full link should be shortened to: https://apps.who.int/iris/bitstream/handle/10665/186463/9789240694811_eng.pdf

Round 2

Reviewer 1 Report

The work is current, however, I believe that there is a relevant number of studies that could be referenced and used to support the work...

The inclusions made by the authors improve the quality of the text and better illustrate the experiments carried out.
